# Optical interferometry based micropipette aspiration provides real-time sub-nanometer spatial resolution

Massimiliano Berardi [1,2✉], Kevin Bielawski[2], Niek Rijnveld[2], Grzegorz Gruca[2], Hilde Aardema[3], Leni van Tol[3], Gijs Wuite[1] & B. Imran Akca [1✉]

Micropipette aspiration (MPA) is an essential tool in mechanobiology; however, its potential is far from fully exploited. The traditional MPA technique has limited temporal and spatial resolution and requires extensive post processing to obtain the mechanical fingerprints of samples. Here, we develop a MPA system that measures pressure and displacement in real time with sub-nanometer resolution thanks to an interferometric readout. This highly sensitive MPA system enables studying the nanoscale behavior of soft biomaterials under tension and their frequency-dependent viscoelastic response.

[1] LaserLab, Department of Physics and Astronomy, VU University, De Boelelaan 1081, Amsterdam, The Netherlands. [2] Optics11, De Boelelaan 1081, Amsterdam, The Netherlands. [3] Department of Farm Animal Health, Faculty of Veterinary medicine, Utrecht University, Yalelaan 7, Utrecht, The Netherlands. ✉email: massimiliano.berardi@optics11.com; b.i.avci@vu.nl

Cell mechanobiology has become increasingly important for investigating and understanding cell physiology and pathology[1–4]. For this reason, several methods have been developed to study single-cell mechanics[1–11]. Among them, MPA is one of the most widely adopted techniques due to its simplicity. In a traditional MPA[5] experiment, a cell is immobilized at the tip of a small glass pipette having internal diameter $R_p$, and a negative pressure is applied to draw the cell into it. The aspirated length ($L_p$) of the cell, subject to an increasing pressure in time, is tracked using an optical microscope. This technique has, however, some major limitations. First, the spatial resolution is defined by the microscope camera, which is limited to hundreds of nanometers even when sub-pixel detection algorithms are employed[12]. Second, the readout is susceptible to drift and lacking plate/capillary parallelism may introduce projection errors. These limits make it practically impossible to measure small deformations[5] ($L_p/R_p < 0.001$), which is important not only for measuring consistent values of the cells' Young moduli but also for investigating the effect of different internal components of the cytoskeleton on cell mechanics[8,11]. Third, the post-processing of microscope images using advanced image processing techniques can be slow, complicated, and computationally demanding, contributing significantly to the total time of a single experiment[12,13]. Although parallel testing approaches based on microfluidics improved MPA throughput[14], the resolution cannot be improved by these methods.

To overcome these limitations, we developed a unique MPA system based on an interferometric all-optical readout approach. We achieved this by assembling a probe (Fig. 1d, Supplementary Figs. 1–3) that consists of a micro-lensed optical fiber inside a glass capillary, pointing at its aperture, and a microelectromechanical system (MEMS)-based fiber-top pressure sensor, located a few millimeters away from the aspiration point. The sample is illuminated through the micro-lensed fiber and the motion of the sample is tracked through the phase variations of the backscattered light from the sample surface, which is then used for extracting sample displacement values. Similarly, the pressure is monitored through the MEMS deformation with the second optical fiber. Using this approach, we obtained real-time, consistent and precise mechanical measurements, featuring sub-nanometer resolution in biologically relevant force ranges. Our approach simplifies data retrieval and eliminates tedious and time-consuming post-processing techniques, without substantially changing the traditional workflow and modeling. Moreover, it expands on the current experimental possibilities and allows for the analysis of the dynamic behavior of soft bodies at the nanoscale.

## Results and discussion

**System design**. The MPA system is comprised of three main parts: an interrogator, a displacement sensor, and a pressure sensor. Figure 1a shows the optical setup of the interrogator (Deltasens, Optics11 B.V.). It uses a polychromatic light source (a superluminescent diode, SLD) centered at 1550 nm to illuminate the aspirated portion of the sample and the pressure sensor membrane through optical fibers. An optical power splitter is used to send the input light to the displacement sensor (Fig. 1b) and the pressure sensor (Fig. 1c). The reflected light beams coming from the sample and the pressure sensor are combined at the same optical splitter and are routed via a circulator to the optical spectrometer. The interrogator setup is analogous to a common-path optical coherence tomography system. In this context, however, it is not used for imaging, but instead its intrinsically high stability and optical phase sensitivity[15] are exploited. A detailed list of the components used to assemble the

setup can be found under Methods-Optical design, and Probe Manufacturing and Assembly.

An all-optical readout scheme based on optical interferometry is implemented in the MPA system. To this end, we analyzed two optical cavities that are formed between (1) the end facet of the sample fiber and the aspirated surface of the sample, and (2) the pressure sensor fiber and the surface of the pressure sensor membrane. These cavities behave like extrinsic, low-finesse, Fabry-Perot resonators and they are of different lengths by design. In this way, the interference pattern of each cavity can be analyzed independently (Fig. 1a, spectrometer inset, and Supplementary Movies 1 and 2) and can be separated by taking the Fourier transform of them. Since the spectrum is evenly sampled in $k$-space, the Fourier transform of the fringe pattern yields a corresponding cavity space. The frequency of such fringes depends on the geometrical size of the cavity and its refractive index (RI); therefore, after applying the Fourier transform, the contribution of different parameters can be separated (Fig. 1e). By demodulating the phase variation of the optical cavities over time, the aspirated length of the sample, i.e. displacement, can be obtained very precisely. This information is retrieved from the point spread function peaks of corresponding cavities in the form of optical path length variation ($\delta$OPL) over time by using[15]:

$$\delta OPL_i(t) = \phi_i(t) \cdot \frac{\bar{\lambda}}{4\pi} = n_i \cdot d_i(t) \qquad (1)$$

where $\phi_i$ is the unwrapped phase at time $t$ for the $i$-th cavity, $\bar{\lambda}$ is the mean wavelength of the source, $n_i$ and and $d_i$ are the RI and the geometrical length variation of the $i$-th cavity, respectively. The RI of the fiber-to-sample surface is that of the medium being used. During testing, its value remains unchanged, allowing for the extrapolation of $d_i$. An example is given in Fig. 1f where the isolated pressure peak is demodulated and its phase is unwrapped. The result is the deflection of the MEMS membrane over time as a response to pressure variations created by the reservoir and the syringe pump. This cavity is sealed, so the medium is air, which makes $\delta OPL_i(t) = d_i(t)$.

The amount of the light that is backscattered from RI discontinuities of the sample depends on the size, geometry and optical properties of the sample (cavity b in Fig. 1c). Since the light source is polychromatic, it is possible to estimate the absolute optical size of the sample and calculate either the geometrical size or the RI of it, if one of these terms is known. Provided that the test is non-destructive and the sample does not undergo structural changes such as strain-induced crystallization, the cavity has a constant average RI during aspiration. As this work focuses on the mechanical characterization, only the point spread function peak of the sample-to-fiber cavity is analyzed. In the Supplementary Information, we provide more insight on the possibilities and limitations of the optical characterization (see in particular Supplementary Fig. 4).

The other unique feature of the MPA system is that the frequency-dependent sample viscoelasticity can be studied through dynamic mechanical analysis (DMA), an unexplored method in traditional MPA systems[16]. This is achieved by connecting the displacement and pressure sensors to the same light source. In this way, the physical excitation and the material response are encoded in the same interferometric signal and therefore a natural synchronization between these parameters is obtained. In a DMA test, a frequency-varying sinusoidal load is imposed on a sample, and the material response is recorded. Owing to its viscous characteristics, the deformation will be out of phase with the excitation. The quantification of this phase lag, the magnitude of elastic/viscous components, and the amount of energy dissipated in each loading cycle provide important

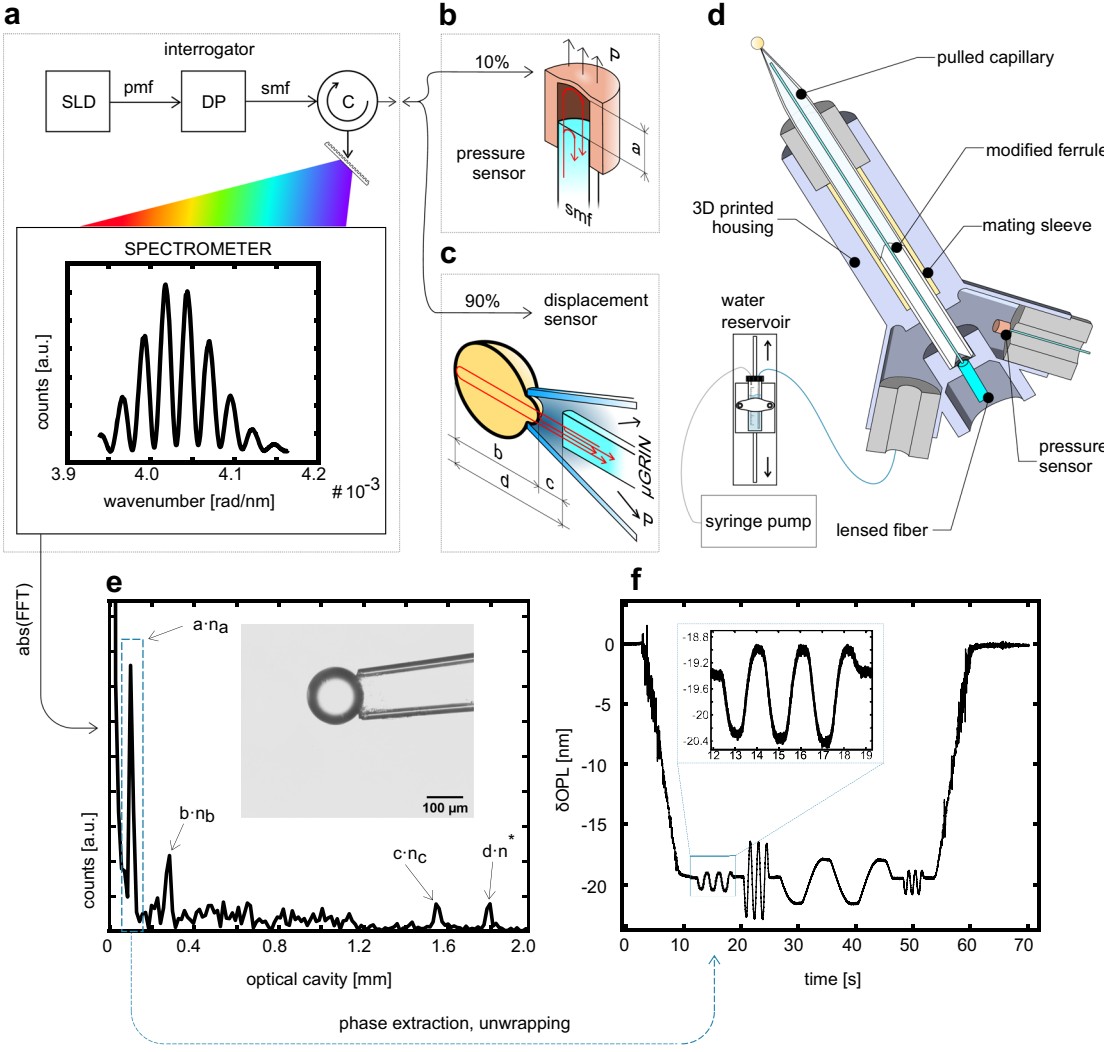

**Fig. 1 Schematics of the optical readout setup and data analysis process. a** The optical interrogator (Deltasens) is comprised of a superluminescent diode (SLD) connected to a depolarizer (DP) via a Panda fiber (pmf). The DP is then connected to a circulator (C) with a single mode fiber (smf), which reroutes the signal to the spectrometer and, via a 90/10 optical fiber splitter, to the sensors. **b, c** These are shown schematically (not to scale) to highlight the Fresnel reflections occurring at RI discontinuities (red arrows), with the resulting optical cavities (dimensions a to d). **d** Schematic cross-section of the probe where its components and the connection to the water reservoir and syringe pump are shown. During an experiment, a pressure variation at the probe nozzle can be operated either by changing the height of the water reservoir or by operating the syringe pump. **e** Fourier transform of an interference signal taken after capturing a 160 μm polystyrene bead (as shown in inset). The $n$ subscripts refer to the RIs of the media between the interfaces, referring to the dimensioning shown in the sensors highlight ($n^*$ is a weighted average of $n_b$ and $n_c$). Each peak contains phase information. By selectively tracking its variation over time, it is possible to obtain optical path length variation, δOPL, versus time plots. **f** Example of the unfiltered pressure sensor response to a DMA-like test. Note the sub-nanometer resolution, highlighted in the inset.

information about the intra- and inter-molecular interactions as well as their variation in time and with temperature[16,17].

**System validation**. We validated the performance of our probe on water-rich alginate microbeads, flying fish roe samples, and matured (metaphase II meiosis) bovine oocytes, following the experimental procedure described in Methods section.

First, we used alginate microbeads (0.5% w/v, diameter ~300 μm, IamFluidics B.V.) to compare our measurement technique to the standard MPA approach[5] (Supplementary Fig. 5). We positioned the probe in between two glass slides, so that the aspiration would be perpendicular to the objective. We recorded a video of the bead displacement inside the capillary during the aspiration portion of the experiment. As Fig. 2a shows, the results of the video tracking and the demodulated phase of the fiber-to-sample peak (once

corrected for the refractive index of water) are in excellent agreement. The samples showed a linear dependency between applied pressure and aspirated length, and a marked viscoelastic behavior. In Fig. 1a, it is possible to see how, starting at ~15 s, the sample undergoes creep (almost 1 μm) without reaching a deformation plateau, while the pressure is held constant for 10 s. We extrapolated the elastic modulus, $E$, by fitting the aspiration portion of the curve, following the procedure described in Methods-MPA validation section. The calculated elastic modulus (mean ± SD, $N = 10$) was $50 \pm 6$ kPa, which agrees well with the value declared by the manufacturer, i.e. $50 \pm 20$ kPa.

We then extended the quasi-static model used on alginate beads and applied it to the DMA testing of the flying fish eggs (see Methods section, Supplementary Figs. 6–8, and Supplementary Table 1) in a range that is not accessible in a standard MPA

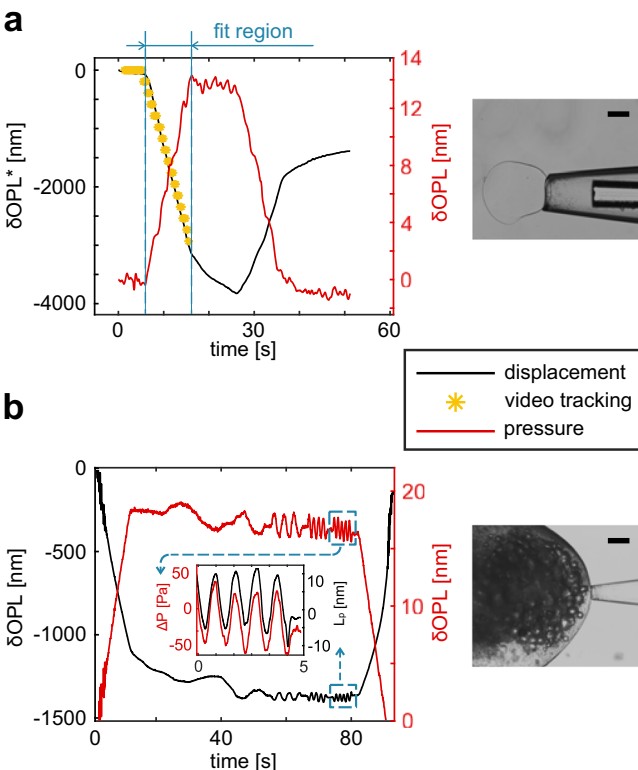

**Fig. 2 System validation. a** Quasi-static aspiration of an alginate bead. The plot shows the variation of the phase of the cavities associated with the pressure sensor (red line) and the displacement sensor (black line). Since a negative pressure is applied, the fiber to sample distance is reducing while the pressure sensor cavity is increasing. To compare the fiber-to-bead cavity variation with the video tracking, the displacement measurement ($\delta OPL^*$) is corrected by dividing it by the refractive index of water ($n = 1.331$). The membrane displacement is represented with yellow asterisks. **b** DMA on a flying fish roe, with a highlight on one of the test frequencies (red line for pressure, black line for aspirated length). Images of the alginate bead and the flying fish roe captured by the measurement probe are shown in the right column. Scale bars are 100 μm.

system. An example of the experimental results is given in Fig. 2b. The elastic modulus of the fish eggs was calculated to be $360 \pm 50$ kPa, with $N = 10$.

Finally, we performed DMA tests on bovine oocytes (Fig. 3a) to assess the frequency-dependent rheological properties of the Zona Pellucida (ZP) at a fixed temperature of 37 °C. The ZP is a thin extracellular matrix that is composed of a randomly oriented filament network of three glycoproteins; ZP2, ZP3, and ZP4[18]. These gylcoproteins undergo large conformational changes during the oocyte maturation, which leads to variation in mechanical properties[4,18]. By applying a very low-pressure oscillation, we can approximate the deformation occurring only on the outer membrane of the oocyte. This allows studying the behavior of the ZP isolated from any other contribution from the underlying structure.

The storage (E') and loss (E'') modulus values at the test frequencies are plotted in Fig. 3b. To highlight the phase lag between the excitation and the material response, we show the time variation of the pressure and the aspirated length at 0.75 Hz in Fig. 3c. Both E' and E'' show a frequency, $\omega$, dependent behavior with values increasing from E'=$39 \pm 6$ kPa and E'' = $6 \pm 5$ kPa at $\omega = 0.05$ Hz to E' = $62 \pm 19$ kPa and E'' = $14 \pm 7$ kPa at $\omega = 1$ Hz. This dependency is commonly observed in biological systems[1,4,7,18] and generally well described by a two-term power

law that accounts for a low-frequency regime with a weak exponential dependency followed by a high-frequency regime where E' and E'' eventually cross each other. It takes the form[19]:

$$E^*(\omega) = A \cdot (i\omega)^\alpha + B \cdot (i\omega)^{3/4} \qquad (2)$$

with $A, B$ and $\alpha$ are the fitting parameters. The best fitting parameters for our sample are $A = 40.50$ kPa, $\alpha = 0.0908$ and $B = 4.32$ kPa. They describe a trend that is within the standard deviation of our measurement results. However, in order to have a more robust model for this behavior, we would need to expand our dataset to cover a broader range of frequencies, which is currently limited by the actuation speed of the syringe pump.

This result is particularly useful to highlight the advantages of our technique: unlike previous works[4,18,20], our measurement is in-situ, does not require sample preparation and could be implemented in standard IVF practices without any workflow change. The small radial strain we measured ($\varepsilon_{n,R} = dR/R_0 \cong 0.5\%$, where $R_0 = 10$ μm is the thickness of the ZP) cannot be investigated with the standard MPA approach. Despite the absence of reports on the dynamical properties of the ZP, we can compare the storage modulus value extracted at the lowest excitation frequency by assuming that it is approaching to the elastic modulus value. Previous studies where the ZP mechanical properties were tested using atomic force microscopy report stiffness values of mature oocytes ranging from $22 \pm 5$ kPa[18] to $32 \pm 9$ kPa[4], which are close to our measurement result of $39 \pm 6$kPa. However, it is important to note that in these AFM studies the ZP was separated from the cytoplasm via vortexing, which may influence its mechanical properties. Moreover, both the scale and testing direction differ, making direct comparison difficult. For the complete set of results, we refer the readers to the Supplementary Figs. 9–11, and Supplementary Table 2.

In conclusion, our interferometry based MPA system substantially improves the spatial (displacement) resolution over state-of-the-art MPA, simplifies data retrieval and opens a new way to study mechanical properties on a wide range of samples. The RI difference between the outer surface of the sample and the sample medium defines the signal strength and contrast. Larger differences in RI will provide more contrast and clearer signal. As we showed by measuring the mechanical properties of hydrogel beads, the system allows for the retrieval of signals from samples that provide lower contrast (RI~1.332[21]) than cells (RI = 1.34–2.2[22,23]) in water. We believe this technique could be particularly useful for certain applications where high precision in membrane positioning and analysis are crucial such as studying lipid vesicles[24] (RI ~ 1.4–2[25–28]) and biomembranes at the nanoscale, as we showed in the case of the ZP.

The modular design of the probe, based mostly on minimally altered off-the-shelf components, allows replacement of the capillary in case of damage, contamination, or clogging while keeping the cost down. It is also compatible with traditional MPA setups, as well as standard capillary manufacturing procedures. Due to the possibility of multiplexing several cavities, multiple probes can be used simultaneously to perform more elaborate experiments, such as cell-cell adhesion studies[29]. The pressure and displacement retrieval method obviates the need to have the pipette tip parallel to the bottom of the sample holder for microscopic imaging, which makes it possible to work directly in well plates. Moreover, because of the placement of the fiber in the capillary, the displacement measurement is drift free.

The probe manufacturing process of this new approach represents a main challenge. Whilst in terms of concentricity we had repeatable results, interfacing four components (fiber, ferrule, 3D-printed housing and capillary) resulted in large

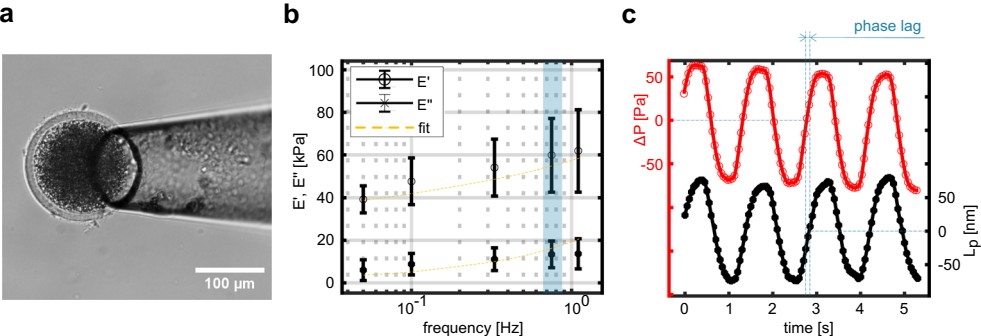

**Fig. 3 DMA of the Zona Pellucida of mature bovine oocytes. a** Image of an oocyte captured by the measurement probe. The thin, semitransparent membrane around the cell is the Zona Pellucida. Its thickness, measured via image analysis, was about 10 μm for all the tested oocytes. It is worth noting that, since the displacement is measured via the fiber within the capillary, the probe does not need to be parallel to the petri dish. **b** The frequency dependency of storage and loss moduli of the Zona Pellucida, (mean ± SD, $N$=10). The solid yellow line is the expected trend as a function of frequency, following the two terms power law behavior mentioned in the main text. **c** Variation of pressure (ΔP) and aspirated length ($L_p$) at 0.75 Hz, highlighting the phase lag between the two signals.

variations in relative axial positioning, which increased the length of the fiber-to-sample cavity (~300 μm at times). This extended cavity length degraded the system sensitivity due to the fact that 1) the optical beam is no longer focused on the sample surface and 2) longer cavities provide less contrast[15]. Similarly, nozzles, which have a diameter smaller than the beam waist, reduce the amount of light reflected from the sample. We were able to retrieve signals from cavities as large as 1.6 mm with pipette nozzles of 20 μm (see Supplementary Movie 2), which means that the technique is currently suitable for studying large bodies, such as oocytes and organoids. This limitation could be solved with a tighter control of the tolerances of the housing and the capillary total length. For example, the glass pulling procedures can produce conical tips with a tapering half-angle of $\theta_t$~8/10 degrees[30], which would allow having $R_p$ = 2.5 μm and a fiber-to-sample cavity length of ~500 μm. Alternatively, it is possible to substitute the pipette with functionally equivalent components that are manufactured using optical lithography[31] or high-resolution 3D printing[32]. In both cases, the goal would be to reduce the length of the fiber-to-sample cavity and make the probe fabrication more reproducible.

As previously mentioned in the discussion, another limitation lies in the small frequency range that can be tested. However, this is a limitation in the actuation speed of the syringe pump piston and it is not related to the detection method, as the sampling frequency can be set to several kHz. This limitation could be solved by changing the syringe with e.g. a piezo-based pump or a rotary aspirator.

Considering the exciting results that we obtained with the optical interferometry-based MPA system, we believe that it will be a game changer in mechanobiology and it will unlock a range of exciting applications from rheological measurements during in vitro fertilization treatment to high-throughput mechanopharmacology studies.

## Methods

**System design and experimental capabilities**. The MPA system includes a syringe pump (neMESYS 290 N, Cetoni), a motorized stage (LTS300/M, Thorlabs), a micromanipulator (PatchStar, Scientifica), and a home-built inverted microscope connected to a camera (Retiga R1, Teledyne QImaging). The components are controlled via a custom-made LabVIEW program. The user can apply pressure on the sample by operating both the reservoir and the syringe, depending on the protocol. For quasi-static and creep tests, the water reservoir is displaced in an arbitrary number of steps defined in pressure and time. The stage has an absolute positioning accuracy of about 50 μm, which in turn means the applied pressure is known with less than 0.5 Pa uncertainty. For DMA tests, the preload is reached by moving the reservoir, whilst the sinusoidal load is obtained by oscillating the syringe plunger. This allows for both isofrequency strain sweeps, to assess linear

viscoelasticity boundaries, and isostrain frequency sweeps, for $E'$, $E''$, and $\tan\delta$ measurements.

**Optical design**. The interrogator uses a broadband light source (SLD, 50 nm FWHM centered at 1550 nm, 21 mW at 500 mA), which is connected to a depolarizer via a polarization maintaining fiber. The output of the depolarizer is sent to a home-built spectrometer (transmission grating based, wavelength range 1510–1595 nm, pixel resolution 166 pm) and the pressure and displacement sensors via an optical circulator (FS.COM GmbH). The output of the interrogator is coupled to a 1 × 2 single mode wideband fiber optic coupler (90/10, TW1550R2A1, Thorlabs). The tap output is used to read the response of the MEMS-based pressure sensor (76.59 Pa/nm, Optics11 B.V.), that is placed in close proximity to the capillary (Fig. 1c). The signal output is used for the displacement readout, with an optical fiber placed within a glass micropipette, pointing at its aperture (Fig. 1d).

**Probe manufacturing and assembly**. For a reliable detection of the displacement, the probe design needs to satisfy three conditions: good fiber and capillary concentricity for optimal light coupling, mechanical stability for a consistent readout, and sufficient water flow. To achieve this, we designed and manufactured a holder (see Fig. 1d) with a desktop stereolithography 3D printer (Form2, Formlabs), using clear resin (FLGPCL02, Formlabs) and 25 μm print layers. To hold the fiber in place, we used a ceramic ferrule (CFX126-10, Thorlabs), on which we carved three ≈300 μm deep slits using a diamond wire cutter. We inserted the modified ferrule in the main channel of the holder, followed by a ceramic mating sleeve (ADAL1, Thorlabs). The geometry of the internal channel makes it so that there is only a partial coupling between the two components, and its length is predictable, which is an important feature for selecting the cavity to be demodulated. The remaining length of the mating sleeve houses the suction capillary, which is held in place by a gasket. Using 1.2 mm OD/0.9 mm ID capillary (Scientific Instruments GmbH), there is a very small backlash between pipette OD and ceramic sleeve ID. This, together with the chamfered end of the ferrule and the compression operated by the silicone sleeve ensured good concentricity. Since the length of the support operated by the mating sleeve is predictable, we can manufacture capillaries with minimal length (see the photo in Fig. s1-a). This is important because as the fiber pokes out of the ferrule, it will inevitably bend due to fiber curl. With the current geometry, we can ensure the fiber bending deformation to be smaller than 10 μm for a curl radius of 4 m, i.e. the lowest acceptable radius in standard manufacturing practice.

We pulled the capillaries with a Narishige SR-10 and cut them to length with respect to the previously described geometrical requirements using a diamond wire cutter. We cut the nozzle to the desired diameter using a ps-laser ablation system (Optec System with Lumera laser source) and inserted the capillary in the main channel of the holder. Since this is not a standard practice in MPA experiments, in the Supplementary Information we discuss the effect of the surface roughness that arises from this different manufacturing route. In particular, we point the reader to Supplementary Fig. 3, where we show a reconstruction of the capillary nozzle along with a roughness measurement.

Since we aimed at measuring water-rich samples, we tried to improve SNR by using fiber-top μGRIN lenses[33,34] to focus the beam, so its waist would be located at the capillary end. We designed the working distance and the beam waist of the μGRIN lens with a MATLAB script based on the ABCD matrix method[35]. The targeted beam waist value was ≈30 μm at 500 μm in air, which was a compromise between the minimum beam waist and the longest working distance that is limited by the tapering capabilities of our pipette puller. Using a glass workstation (LDS 2.5, 3SAE), we spliced a 9/125 μm single mode optical fiber (Corning ClearCurve

ZBL, equipped with E2000 connector) to a 300 μm long coreless fiber (FG125LA, Thorlabs), and later to a 50/125 μm, 498-μm-long GRIN multimode fiber (GIF50C, Thorlabs). We characterized the probe with a scanning slit beam profiler (BP209-IR2/M, Thorlabs). The results are given in Supplementary Fig. 1c, d. We glued a short portion of tubing at the end of the stripped portion of the fiber and inserted in the ferrule through the small back opening in the holder. The tubing provides a watertight fit, and since no glue is applied to the ferrule, the fiber can be replaced if damaged.

Lastly, we inserted two short portions of tubing in the lateral openings of the holder. The first tubing was then connected to the water reservoir using a standard tube connector (ISM 556, Ismatec), whilst the second was used as a sleeve to insert the pressure sensor (Supplementary Fig. 1b). Given the pressure that can be exerted by moving the reservoir (up to 3 kPa), interference fits were enough to guarantee a watertight sealing. A picture of one of the fully assembled probes is available in Supplementary Fig. 1a.

**MPA validation and data analysis**. We placed a small amount of distilled water in between two glass slides, relying on surface tension to keep the sample in place. We filled the water reservoir with distilled water and by using the syringe pump the probe was filled without any air bubbles trapped in the system. We then adjusted the pressure in the reservoir as well as its height, until we could not observe any flow through the pipette. As the micropipette manufacturing method is not conventional, we took additional steps to ensure the process allows for conformal contact between sample and nozzle. We placed the reservoir above the pipette, which results in a positive pressure at the nozzle, and we counterbalanced this using the syringe pump. If the seal is proper, the pressure becomes stable when the sample is captured. This is verified by observing the motion of debris in proximity of the nozzle and the demodulated phase of the pressure sensor. During an experiment, we applied a gentle negative pressure (~100 Pa) to capture a bead. Then, we applied a trapezoidal suction profile, starting with a ramp of 100 Pa/s, a wait period of 10 s at peak pressure (1 kPa), and a symmetric release. Simultaneously, we recorded the interferogram of the entire procedure at 1 kHz and a time-lapse of the membrane undergoing aspiration at 7 fps. We intentionally oversampled the interferogram to improve the SNR during post processing, by means of filtering (see Methods-Data Analysis). For this experiment, we used capillaries with a diameter of 90 μm and 150 μm.

We extrapolated the elastic modulus with a linearized version of the Zhou model[8], as presented by Plaza et al.[36], to account for the finite size of each bead:

$$\frac{\triangle P}{E} = \frac{\beta_1 \left[1 - \left(R_p/R_c\right)^{\beta_3}\right]}{3} \frac{L_p}{R_p} \quad (3)$$

where $\Delta P$ is the differential pressure, E is elastic modulus, $L_p$ is the aspirated length, $R_p$ is the pipette radius, $R_c$ is the radius of the aspirated body measured via brightfield microscopy, $\beta_1 = 2.0142$, and $\beta_3 = 2.1187$. The sample is assumed to be incompressible (ν = 0.5). We placed the flying fish roe in the same holder used for the alginate beads, and followed the same preparation described earlier. The hysteresis curves were acquired by programming a triangular pressure profile (1500 Pa peak pressure, 150 Pa/s, symmetrical loading/unloading, repeated 5 times), operated by the motorized reservoir. The DMA tests were performed by applying a preload of 1500 Pa, a wait time of 10 s followed by 5 oscillatory periods (60 Pa amplitude, testing frequencies 0.05 Hz 0.1 Hz, 0.35 Hz, 0.75 Hz, and 1 Hz) separated by 2 s of rest.

The viscoelastic parameters storage E' and loss moduli E" were extrapolated by extending the Zhou and Plaza model described earlier to analyze the dynamic behavior of materials. Assuming a sinusoidal input of angular frequency ω, the pressure and aspirated length can be described as a function of time given as:

$$P(t) = P_0 \sin\left(\omega t + \delta_P\right) \quad (4)$$

$$L_p(t) = L_0 \sin\left(\omega t + \delta_L\right) \quad (5)$$

Where $P_0$ and $L_0$ are the amplitudes of pressure and aspirated length oscillations, and $\delta_P$ and $\delta_L$ are their corresponding phase shifts, which are non-zero and generally different from each other. Using basic trigonometric rules, we can rewrite (4) as:

$$L_p(t) = L_0 \left[\sin(\omega t)\cos\left(\delta_L\right) + \cos(\omega t)\sin\left(\delta_L\right)\right] \quad (6)$$

From which it appears clear that a general viscoelastic material response is comprised of two elements: one term in phase with the input, and one at the quadrature. For dynamic analysis, it is particularly convenient to express these terms as a complex number by rewriting the trigonometric quantities in their exponential notation. Inserting (3) and (4) into (2) yields:

$$E^* = \frac{3R_p}{\beta_1[1-(R_p/R_c)^{\beta_3}]} \cdot \frac{P_0 e^{i(\omega t + \delta_P)}}{L_0 e^{i(\omega t + \delta_L)}}$$
$$= \frac{3R_p}{\beta_1[1-(R_p/R_c)^{\beta_3}]} \cdot \frac{P_0}{L_0} \cdot e^{i(\delta_P - \delta_L)} \quad (7)$$

Where $E^*$ is called complex modulus. Knowing that in the limit of $\omega \to 0$ $L_p$ will be in phase with the input (perfectly elastic response), we can define storage and

loss moduli as:

$$E' = \frac{3R_p}{\beta_1 \left[1 - \left(R_p/R_c\right)^{\beta_3}\right]} \cdot \frac{P_0}{L_0} \cdot \cos\left(\delta\right) \quad (8)$$

$$E'' = \frac{3R_p}{\beta_1 \left[1 - \left(R_p/R_c\right)^{\beta_3}\right]} \cdot \frac{P_0}{L_0} \cdot \sin(\delta) \quad (9)$$

where $0 \le \delta = \delta_P - \delta_L \le \pi/2$ represents the phase lag between input load and material response. During fitting, we also included a linear correction factor for pressure and displacement, to account for incomplete sample creep and to perform a more accurate amplitude/frequency fit.

**Oocytes preparation and testing**. Bovine ovaries were collected from a local abattoir and transported to the laboratory within 2 h after withdrawal. Ovaries were washed in physiological saline (0.9% NaCl) and kept in physiological saline with 0.1% (v/v) penicillin-streptomycin (Gibco BRL, Paisley, U.K.) at a temperature of 30 °C. Follicles ranging from 3 to 8 mm were aspirated under low vacuum by a suction pump with a 19-gauge needle and allocated to a 50-ml conical tube. Cumulus oocyte complexes (COCs) with a minimum of three layers of cumulus cells were selected and first washed in HEPES-buffered M199 (Gibco BRL) and subsequently washed and cultured in M199 maturation medium (Gibco BRL) supplemented with 2.2 mg/ml NaHCO₃. Selected COCs were cultured in four-well culture plates (Nunc A/S, Roskilde, Denmark) containing maturation medium M199 (Gibco BRL) supplemented with 0.05 IU FSH/mL (Organon, Oss, The Netherlands), 0,1 μM cysteamine and 10 ng/mL EGF and 1% (v/v) penicillin-streptomycin (Gibco BRL). The oocytes were matured in groups of 35 COCs in 500 μl and incubated under a humidified atmosphere of 5% CO₂ in air for 23 h at 39 °C. They were then moved from the falcon tube to a 60 mm petri dish using a micropipette. A total of 10 Oocytes were tested, at 37 °C. Temperature control was achieved by enclosing the setup in an isolation box and by adding a temperature control system that reads the medium temperature with a thermocouple (see Supplementary Fig. 5). Each oocyte was manually captured by applying a small negative pressure of about 50 Pa. Once the sample stopped creeping in the pipette, we applied the test protocol; a load ramp of 50 Pa, 4 s of wait, and a set of oscillations analogous to what was used for the flying fish roe, with ~30/50 Pa amplitude. The moduli displayed in the plots are calculated under the assumption of the ZP as an infinite half-space. This can be obtained using the model previously described, in the limit $R_p/R_c \to 0$. This is motivated by the fact that the thickness of the ZP is much larger than the applied deformation.

**Statistics and reproducibility**. The elastic and complex moduli were extrapolated using a custom MATLAB script (Supplementary code) that implements the models described in the main text and in the Methods section. Since the samples under study (and soft biological bodies in general) are overdamped, we can assume that their frequency-specific response depends solely on external excitation. This justifies filtering high frequency noise recorded in the phases of the FP cavities. The demodulated data was filtered using a third order, low-pass finite impulse response Butterworth filter, to ensure maximally flat response in the passband window. The cutoff frequency of 25 Hz was selected by inspecting the periodogram of the pressure sensor and defined as the point where power and frequency reaches −40 dB/Hz. The values are displayed as sample mean ± standard deviation, with a sample size of 10, without repeated measurements on the same body (either hydrogel, flying fish roe, or oocyte).

**Reporting summary**. Further information on research design is available in the Nature Research Reporting Summary linked to this article.

## Data availability
All raw and processed data that support the findings of this study are given in ref. [37]. Any remaining information can be obtained from the corresponding author upon reasonable request.

## Code availability
Supplementary code is given in ref. [38].

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

## Acknowledgements

This work was financially supported by the H2020 European Research and Innovation Programme under the Marie Skłodowska-Curie grant agreement "Phys2BioMed" contract no. 812772. The authors would like to thank Stefan Werzinger and Noor Schilder for providing parts of the MATLAB code used to model the μGRIN lenses, Lily Kardomateas for help in manufacturing pressure sensors, and Prof. Davide Iannuzzi for being the major driving force behind this work.

## Author contributions

M.B. built the MPA system, designed and fabricated the probe, performed the experiments, and wrote the manuscript. K.B., G.G., N.R. and B.I.A. assisted in design, and processing of the experimental data. M.B., G.G., N.R. and B.I.A. designed and planned the project. L.T. and H.A. prepared and provided the oocytes. B.I.A., G.W., K.B. and H.A. contributed to the manuscript.

## Competing interests

The authors declare the following competing interests: M.B., K.B., N.R., and G.G. are employed at Optics11. The remaining authors declare no competing interests. A patent describing the optical system and a data acquisition/analysis procedure has been awarded to Optics11 (WO2017077138A1).
