## [Peer Review File · Communications Biology]

Reviewers' comments:

Reviewer #1 (Remarks to the Author):

Review: Optical Interferometry Based Micropipette Aspiration with Real-time Sub-nanometer Spatial Resolution

Brief Summary and Overview:

In this article the authors significantly enhance the sensitivity and performance of micropipette aspiration technique by integrating it with optical interferometry and MEMS based pressure sensor. This all-optical readout combination allows them to measure pressure and the aspired length with a sub-nanometer resolution in real time. This allows for the characterization of the elasticity of cells and materials at very small deformations, yielding consistent and reproducible values. Besides, this unique setup enables studying frequency-dependent rheology of materials. The setup's performance is validated on three samples, alginate microbeads, bovine oocytes and flying fish roe, and the viscoelasticity measurements are carried out on bovine oocytes. This work is mostly technical and will definitely lead to significant future work and methodology in the field of mechanobiology. The results clearly meet the requirements and the goals set in the work. There are a few concerns and recommendations that can improve the manuscript in its current state and warrant its publication in *Communications Biology*:

1. The authors introduce several abbreviations that are sometimes not used more than once or twice in the manuscript, such as LVE, SLD, DP, PS, etc. This makes readability a bit difficult. There are certain abbreviations that are not introduced such as SLA. Could these abbreviations be avoided in such cases?
2. The results are presented but not appropriately discussed in the article. A few points such as comparison with the traditional MPA values, or in the case of viscoelastic moduli a comparison with other techniques such as AFM based microrheology could be useful for the readers. In this regard, was a traditional MPA experiment performed on the same cells to comment and compare the elasticity results? This will significantly highlight the advantages and maybe bring out any limitations of the current methodology.
3. Decoupling refractive index and the geometric length that enter multiplicatively in the optical phase difference relationship is a long standing problem. It is more relevant in cell mechanobiology studies that the authors wish to address in this article with this technique. Quoting the article, the authors write "Moreover, the approximate average RI value of the cell can be calculated, if the dimension of the cell is known". This statement sounds ambiguous. A significant advantage of coupling interferometry with the micropipette is the ability to perform experiments with multiple wavelengths. Can the authors comment further? Can a dual-wavelength interferometry be performed to get an approximate estimate of the cell dimensions? Is there a particular reason why the authors choose the wavelength of 1550 nm for these experiments?
4. Can the authors comment on the effect of the applied negative pressure on the cellular cytoskeleton in their experiments with the oocytes?
5. Upon applying a constant pressure, do the authors observe fluctuations in the cellular dimensions? Can the authors comment on the temporal behavior of the cells' viscoelastic properties for a constant load?
6. I think Figure 1-d can be improved to give a better impression of the probe geometry. Probably it can be replaced with the figure in the supplementary information. This would help in better readability and easier understanding the methods section in main article.
7. Figure 2-a and 2-b, the y-axis label is missing. Figure citations in the main text need a revision. Some figure citations such as 1-ii, 2-v/vi do not exist.
8. The variables P_0 and L_0 are not introduced in the main text. Was the radius of the aspired body, R_c determined experimentally from the brightfield image?

Reviewer #2 (Remarks to the Author):

I consider that this is a great work. This new technique provides a huge improvement of the micropipette-aspiration technique and open new possibilities in the cell-mechanics field and beyond.

I think that, formally, the style of the text doesn't follow the typical style of scientific articles, since the main text contains relatively few details describing the results and the discussion is very short. But I think it is very clear and describes well the possibilities of the technique.

Some minor corrections would be needed:

- Caption of figure 1: (vii)  (g).
- Lines 95-98. It is necessary to explain why "Both E' and E'' show a power-law rheological behavior".
- Line 99-100: It would be convenient to include in the text the numerical values of ZP elastic modulus (stiffness) to better understand the explanation.
- Line 153: Figure 1-i  Figure 1a.
- Line 154: Correct "Figure 1-ii", "Fig. 2-v/vi".
- Line 156-157: Confusion using letters for the plots and letters inside plot of Figure 1f.
- Line 218: Ref. number 25  Ref. number 24.

Congratulations to the authors,
Gustavo R. Plaza

Reviewer #3 (Remarks to the Author):

This work introduces an instrumental development representing a fine approach for assessing the mechanics and viscoelasticity of soft beads and large cells. The approach is based on a home-built system, which employs interferometric readout for detecting object displacement (or deformation) as a function of applied pressure. The manipulated objects (beads or cells) are aspirated in microcapillaries via the application of a suction pressure. Their response in terms of membrane or surface displacement is then resolved with sub-nanometer precision. The setup also allows frequency-dependent measurements to assess viscoelastic properties of the samples. The approach is potentially very useful and represents a novel development. However, in its current presentation it is not suitable for publication. Several issues need to be addressed as explained below.

The manuscript is difficult to read. It might be helpful to explain the principle of the setup in layman terms (especially considering the heterogeneous audience of this journal) and in the supplement to present in detail the instrumental aspects and realization. It would be particularly useful if the figure captions (especially those in the supplement) contain more detail. In the current version, the referral to some of the figures is very confusing probably because of a previous version, in which the panels might have been labeled with roman numbers and not with letters.

It would be very interesting if the authors discuss (if not able to demonstrate) the applicability of the approach to investigating vesicles (lipid or polymer giant vesicles). The micropipette aspiration technique has been extensively used in the past (since the 80's) and continues to be applied currently for characterizing the mechanical properties (bending and stretching elasticity and lysis tension) of model lipid and polymer membranes as well as their adhesion, see e.g. the seminal work of Evans (Evans, *Biophys. J.* 31, 425, 1980; Evans & Rawicz, *Phys. Rev. Lett.* 64, 2094, 1990) and more recent studies requiring high precision of membrane position (e.g. Bhatia et al. *ACS Nano* 12, 4478, 2018). The abstract states that the approach is applicable to study biomembranes, which suggests that also lipid vesicles can be probed but since this is not (yet) demonstrated in this work, I would suggest rephrasing the abstract.

Overall, the authors should include specific information about the limitations of the method (maybe in the concluding section). For example, they should specify the optimal range of bead and cell size, at which the approach works best.

How important is and what justifies the assumption that the refractive index of the fiber to the cell back and front surfaces is unchanged?

The authors should introduce a description of how the "standard approach" of manual tracking the membrane displacement has been conducted here. This can be done in the supplement and respective images and data at different suction pressures provided.

The capillaries appear to have not been coated, which is a common procedure to avoid adhesion of cells or model membranes. The authors should demonstrate that for the measured samples this

does not represent an issue. This could be potentially illustrated with tests for hysteresis upon increasing and reducing the suction pressure.

The capillaries (which one clearly sees in the supporting movie) have very rough tip edge, which would suggest typically poor sealing (this could potentially be improved when using a microforge to smoothen the tip). It is thus questionable how easy it is to establish a proper seal during aspiration. Indeed, in this work, proper seal has been verified by monitoring the motion of debris in the proximity of the nozzle. Is the presence of such debris then a necessary condition and doesn't this impose limitations to the approach applicability?

Typos (not a full list) and comments:

Line 59 – “the” is repeated.

Line 155 refers to a measurement with a polystyrene bead in “Fig. 2-v/vi”. This contradicts with the figure caption. First, there are no panels v and vi, and second, the shown bead is an alginate one.

Line 271 – what is the meaning of “thickness of the ZP”? ZP appears to be an abbreviation of Zhou/Plaza.

Response to Reviewers' comments

We thank the Reviewers for the insightful and helpful comments on our manuscript (COMMSBIO-20-2981-T). Below we address each of the Reviewers' comments in turn (in red text). The corresponding changes that we have made to the manuscript are also highlighted in red text. We believe that these changes have greatly strengthened both the clarity and impact of our manuscript.

Reviewer #1 Comment:

In this article the authors significantly enhance the sensitivity and performance of micropipette aspiration technique by integrating it with optical interferometry and MEMS based pressure sensor. This all-optical readout combination allows them to measure pressure and the aspired length with a sub-nanometer resolution in real time. This allows for the characterization of the elasticity of cells and materials at very small deformations, yielding consistent and reproducible values. Besides, this unique setup enables studying frequency-dependent rheology of materials. The setup's performance is validated on three samples, alginate microbeads, bovine oocytes and flying fish roe, and the viscoelasticity measurements are carried out on bovine oocytes. This work is mostly technical and will definitely lead to significant future work and methodology in the field of mechanobiology. The results clearly meet the requirements and the goals set in the work.

There are a few concerns and recommendations that can improve the manuscript in its current state and warrant its publication in Communications Biology:

Detailed Remarks:

1) The authors introduce several abbreviations that are sometimes not used more than once or twice in the manuscript, such as LVE, SLD, DP, PS, etc. This makes readability a bit difficult. There are certain abbreviations that are not introduced such as SLA. Could these abbreviations be avoided in such cases?

As suggested by the reviewer we reduced the number of abbreviations contained in the main text, with one exception: we left the ones in Figure 1-a. to avoid cluttering the image.

2) The results are presented but not appropriately discussed in the article. A few points such as comparison with the traditional MPA values, or in the case of viscoelastic moduli a comparison with other techniques such as AFM based microrheology could be useful for the readers. In this regard, was a traditional MPA experiment performed on the same cells to comment and compare the elasticity results? This will significantly highlight the advantages and maybe bring out any limitations of the current methodology.

Indeed, there was limited discussion in our original submission. We have significantly extended the results section in our revised version as requested by the reviewer.

For comparisons with other studies, traditional MPA cannot be used in the range we are investigating (please see answer 4 below). We found two articles (Refs. 4 and 9 in the main text of the revised manuscript) that use AFM to assess the mechanical properties of the Zona Pellucida of bovine oocytes in the same deformation range. The presented values can be compared to our lowest frequency measurement, under the assumption of negligible viscous contribution. The values are quite close (22-32 kPa vs 39 kPa). Nevertheless, we believe that a direct comparison is quite difficult because of the following reasons:

- 1) We manufactured the pipette tips using laser ablation, which resulted in a roughness ($R_q \sim 100/150$ nm) that is higher than what would be obtained by fire polishing the tip with a microforge. The different contact may change the boundary condition of sliding that the theory requires. For instance, if the membrane was clamped at the nozzle, we would overestimate the mechanical properties. It is likely that the optimum condition is somewhere in between, but measuring it would be extremely challenging. We modelled the Zona Pellucida assuming the boundary effects to be minimum, if present, as we are measuring very small radial displacements at the center of the nozzle that is several microns away from the contact area.
- 2) The AFM tests were performed on the Zona Pellucida after separating it from the cytoplasm. In particular, first, the Zona Pellucida was separated via vortexing, then it was aspirated in narrow bore pipettes, and finally it was attached to glass slides using poly-L-lysine. This procedure applies significant strain on the structure, which could have influenced its structural features.
- 3) The loading mode is significantly different. Both references use sharp tips (a few nm radius), and impose compressive load. Our method, on the other hand, tests a microscopic patch of the material under tension.
- 4) Considering the fact that the structure of the Zona Pellucida is porous at the nanoscale (see *Reprod Dom Anim* 43, 685–689 (2008); doi: 10.1111/j.1439-0531.2007.00970.x), the discrepancy between measurements can be related to the different scales involved and the effects of structural assembly.

Testing the full oocyte, on the other hand, is particularly challenging, as discussed in Reference 21. One would need to either fix them on a surface, quantify the contact area and account for the compliance of the extracellular matrix or partially submerge the sample in a stiff matrix (e.g. agarose), without causing significant pre-strain on the surface to be probed. This is an advantage of our technique as in this scenario our method is far easier to implement.

3) Decoupling refractive index and the geometric length that enter multiplicatively in the optical phase difference relationship is a long standing problem. It is more relevant in cell mechanobiology studies that the authors wish to address in this article with this technique. Quoting the article, the authors write “Moreover, the approximate average RI value of the cell can be calculated, if the dimension of the cell is known”. This statement sounds ambiguous. A significant advantage of coupling interferometry with the micropipette is the ability to perform experiments with multiple wavelengths. Can the authors comment further? Can a dual-wavelength interferometry be performed to get an approximate estimate of the cell dimensions? Is there a particular reason why the authors choose the wavelength of 1550 nm for these experiments?

In order to address these good remarks, we extended the statement in the main text. We chose 1550 nm because of the wide availability and low cost of the components (SLD, spectrometer, fibres). Moreover, the concept can be translated to any wavelength range. Since we are using a polychromatic source, we can estimate the presence of multiple optical cavities based on the different interference patterns they will give rise to. By Fourier transforming the interferogram, we obtain a set of peaks whose absolute position is the geometrical length times the refractive index. Practically, retrieving peaks beyond the sample surface is very challenging in the current iteration of our probe. The reason is limited assembly tolerance; we were rarely able to assemble a probe with a nozzle located at the beam waist. This causes suboptimal light collection, with negative repercussions on the visibility of optical cavities beyond the fiber-to-sample one. Since this is not the main goal of this manuscript, we added a new paragraph and a video in the supplementary information, where we show examples of how the sample cavity visibility

depends on geometry, size, and contrast with respect to the medium. In particular, we show how we can estimate the average refractive index of an oocyte by isolating its peak and measuring its diameter via bright field microscopy.

4) Can the authors comment on the effect of the applied negative pressure on the cellular cytoskeleton in their experiments with the oocytes?

An oocyte is a core-shell structure with two distinct sets of mechanical properties. In order to isolate the contribution of the extracellular matrix and observe only the contribution of the Zona Pellucida, we applied a very low suction pressure and limit the deformation to this area. In this sense, we made an assumption similar to what is used in AFM when studying finite objects with Hertzian contact theory (see <https://www.sciencedirect-com.vu-nl.idm.oclc.org/science/article/pii/S0006349502756208>), where the model is deemed applicable if the surface strain is <10%. Including the viscous contribution, our radial strain is about 2% (0.5% during the DMA phase). We changed the main text to clarify this point

5) Upon applying a constant pressure, do the authors observe fluctuations in the cellular dimensions? Can the authors comment on the temporal behavior of the cells' viscoelastic properties for a constant load?

We do observe fluctuations in the extracellular matrix (ECM) (albeit it is likely isolated to the ECM). Even though we did not model such behaviour, it is possible to see the viscous contribution in Figure s7. The tests begin at pressure 0 Pa and L_p -150 nm, with positive variation in L_p indicated increase in aspirated length. The Lissajous curves produced by the DMA test tend to shift to the right whilst remaining confined between ~40 kPa and ~100 kPa, indicating creep deformation. We extended the caption of the Supplementary figure to clarify these points.

6) I think Figure 1-d can be improved to give a better impression of the probe geometry. Probably it can be replaced with the figure in the supplementary information. This would help in better readability and easier understanding the methods section in main article.

We thank the reviewer for the helpful suggestion. Fig.1d in the revised manuscript was improved accordingly.

7) Figure 2-a and 2-b, the y-axis label is missing. Figure citations in the main text need a revision. Some figure citations such as 1-ii, 2-v/vi do not exist.

Thank you for pointing out the mistakes. We modified the figures and their mentions to improve clarity.

8) The variables P_0 and L_0 are not introduced in the main text. Was the radius of the aspired body, R_c determined experimentally from the brightfield image?

We added their definition after equation (3-4). R_c was determined experimentally, from the bright field image. We added a sentence in the Methods section to clarify this.

Reviewer #2

Comments:

I consider that this is a great work. This new technique provides a huge improvement of the micropipette-aspiration technique and open new possibilities in the cell-mechanics field and beyond.

Detailed Remarks:

1) I think that, formally, the style of the text doesn't follow the typical style of scientific articles, since the main text contains relatively few details describing the results and the discussion is very short. But I think it is very clear and describes well the possibilities of the technique.

This was the result of a manuscript transfer between journals, starting from a brief communication format. We significantly extended the discussion of the results.

2) Some minor corrections would be needed:

- Caption of figure 1: (vii)  (g).
- Lines 95-98. It is necessary to explain why "Both E' and E'' show a power-law rheological behavior".
- Line 99-100: It would be convenient to include in the text the numerical values of ZP elastic modulus (stiffness) to better understand the explanation.
- Line 153: Figure 1-i  Figure 1a.
- Line 154: Correct "Figure 1-ii", "Fig. 2-v/vi".
- Line 156-157: Confusion using letters for the plots and letters inside plot of Figure 1f.
- Line 218: Ref. number 25  Ref. number 24.

We thank the reviewer for providing this list. We corrected the typos and included the numerical values of the Zona Pellucida as well as a more detailed description of its frequency-dependent behaviour.

Reviewer #3

Comments:

This work introduces an instrumental development representing a fine approach for assessing the mechanics and viscoelasticity of soft beads and large cells. The approach is based on a home-built system, which employs interferometric readout for detecting object displacement (or deformation) as a function of applied pressure. The manipulated objects (beads or cells) are aspirated in microcapillaries via the application of a suction pressure. Their response in terms of membrane or surface displacement is then resolved with sub-nanometer precision. The setup also allows frequency-dependent measurements to assess viscoelastic properties of the samples. The approach is potentially very useful and represents a novel development. However, in its current presentation it is not suitable for publication. Several issues need to be addressed as explained below.

Specific Questions:

1) The manuscript is difficult to read. It might be helpful to explain the principle of the setup in layman terms (especially considering the heterogeneous audience of this journal) and in the supplement to present in detail the instrumental aspects and realization.

We significantly extended the discussion of the setup by adding sentences to make the processing steps more explicit and easier to understand. We hope that the current version of our manuscript is now similar, in terms of technical language usage, to other articles published on this journal with a focus on optics.

2) It would be particularly useful if the figure captions (especially those in the supplement) contain more detail.

We extended most of the captions to make the content of the figures easier to interpret.

3) In the current version, the referral to some of the figures is very confusing probably because of a previous version, in which the panels might have been labeled with roman numbers and not with letters.

Thank you for pointing out these mistakes. We modified the figures and their mentions to improve clarity.

4) It would be very interesting if the authors discuss (if not able to demonstrate) the applicability of the approach to investigating vesicles (lipid or polymer giant vesicles). The micropipette aspiration technique has been extensively used in the past (since the 80's) and continues to be applied currently for characterizing the mechanical properties (bending and stretching elasticity and lysis tension) of model lipid and polymer membranes as well as their adhesion, see e.g. the seminal work of Evans (Evans, *Biophys. J.* 31, 425, 1980; Evans & Rawicz, *Phys. Rev. Lett.* 64, 2094, 1990) and more recent studies requiring high precision of membrane position (e.g. Bhatia et al. *ACS Nano* 12, 4478, 2018). The abstract states that the approach is applicable to study biomembranes, which suggests that also lipid vesicles can be probed but since this is not (yet) demonstrated in this work, I would suggest rephrasing the abstract.

Thank you for the comment. We have removed the text related to biomembrane study from the abstract. While we do not have the tools to perform studies on lipid vesicles, we can assume our technique would be applicable to study them, albeit with some limitations in terms of size (the smallest capillary we tested is 20 μm). In the revised Conclusions part, we now list a number of references (22-29) that measure the refractive index of a number of biological structures including lipid vesicles. The fundamental criterion that makes the measurement possible is the refractive index contrast in water. Considering the fact that we have measured samples with RI of 1.331, it is very feasible to measure lipid vesicles (RI \sim 1.4 to 2) with our technique.

5) Overall, the authors should include specific information about the limitations of the method (maybe in the concluding section). For example, they should specify the optimal range of bead and cell size, at which the approach works best.

As suggested, we significantly extended the Conclusion part. As of now, we believe the main limitation of the technique lies in the components assembly and the lack of optimal light back coupling (please see answer 3, reviewer 1). The smallest capillaries we tested were 20 μm in diameter, and the largest optical cavity that allowed retrieving any signal was about 2000 μm .

6) The authors should introduce a description of how the “standard approach” of manual tracking the membrane displacement has been conducted here. This can be done in the supplement and respective images and data at different suction pressures provided.

As suggested, we added a figure (Figure s5) that details how we defined and measured the membrane displacement via image analysis

7) The capillaries appear to have not been coated, which is a common procedure to avoid adhesion of cells or model membranes. The authors should demonstrate that for the measured samples this does not represent an issue. This could be potentially illustrated with tests for hysteresis upon increasing and reducing the suction pressure.

The uniqueness of our MPA system comes from the fact that it can measure very small deformations by aspirating the samples with very low pressures. This also means that the contact between the sample and the inner walls of the capillary is minimum, and thereby the cell adhesion. During our initial experiments, we coated the capillaries with Fetal Bovine Serum, and did not observe a significant difference between coated and uncoated capillaries in terms of adhesion. This is also probably due to

the surface finish and the relatively short contact time. Regarding the proposed hysteresis test, we point to the supplementary Figure s8, where we applied a triangular loading/unloading profile on a fish roe. As expected from a material with a strong viscoelastic characteristic, repeated loading/unloading does not result in the same path repeatedly. Instead, what we observe is a Mullins-like effect (see for example <https://doi.org/10.1016/j.ijsostr.2008.12.015>). In this sense, such a test would be inconclusive.

8) The capillaries (which one clearly sees in the supporting movie) have very rough tip edge, which would suggest typically poor sealing (this could potentially be improved when using a microforge to smoothen the tip). It is thus questionable how easy it is to establish a proper seal during aspiration. Indeed, in this work, proper seal has been verified by monitoring the motion of debris in the proximity of the nozzle. Is the presence of such debris then a necessary condition and doesn't this impose limitations to the approach applicability?

Indeed, the method we used to monitor the quality of the seal is far from optimal. We had to follow this route because we do not own a microforge. We measured via optical profiling (Veeco WYKO NT9100) the surface roughness of some nozzles manufactured according to the method presented in the manuscript, and we found an average roughness of $R_a \approx 100$ nm / $R_q \approx 150$ nm. Since this work was primarily meant to be a proof of concept for the method, we deemed it acceptable. As we mention in the conclusion, the probe can house capillaries that are manufactured with the conventional method (pulling+polishing). This would significantly improve the quality of the contact and ensure a proper seal, as shown in the References 3 and 5 in the main text.

We hope that our careful consideration of the Reviewers' comments will satisfy the Reviewers.

Yours Sincerely,

The authors

REVIEWERS' COMMENTS:

Reviewer #1 (Remarks to the Author):

The authors have addressed all issues raised previously and the manuscript in the current form warrants publication in Nature Communications Biology.

Reviewer #3 (Remarks to the Author):

The authors have addressed all my comments appropriately. One last remark: the fonts of the labels in Fig. 1 are extremely small.